# *Isocitrate Dehydrogenase Alpha-1* Modulates Lifespan and Oxidative Stress Tolerance in *Caenorhabditis elegans*

**DOI:** 10.3390/ijms24010612

**Published:** 2022-12-29

**Authors:** Zhi-Han Lin, Shun-Ya Chang, Wen-Chi Shen, Yen-Hung Lin, Chiu-Lun Shen, Sin-Bo Liao, Yu-Chun Liu, Chang-Shi Chen, Tsui-Ting Ching, Horng-Dar Wang

**Affiliations:** 1Institute of Biotechnology, National Tsing Hua University, Hsinchu 300044, Taiwan; 2Institute of Chemistry, Academia Sinica, Taipei 115201, Taiwan; 3Chemical Biology and Molecular Biophysics Program, Taiwan International Graduate Program, Academia Sinica, Taipei 115201, Taiwan; 4Institute of Biochemical Sciences, National Taiwan University, Taipei 106319, Taiwan; 5Department of Life Science, National Tsing Hua University, Hsinchu 300044, Taiwan; 6Institute of Biomedical Sciences, Academia Sinica, Taipei 11529, Taiwan; 7Institute of Molecular Medicine, College of Medicine, National Taiwan University, Taipei 10002, Taiwan; 8Department of Biochemistry and Molecular Biology, College of Medicine, National Cheng Kung University, Tainan 70101, Taiwan; 9Institute of Biopharmaceutical Sciences, National Yang Ming Chiao Tung University, Taipei 112304, Taiwan; 10Institute of Systems Neuroscience, National Tsing Hua University, Hsinchu 300044, Taiwan

**Keywords:** longevity, oxidative stress, *idha-1*, TCA cycle, dietary restriction, TOR pathway, *C. elegans*

## Abstract

Altered metabolism is a hallmark of aging. The tricarboxylic acid cycle (TCA cycle) is an essential metabolic pathway and plays an important role in lifespan regulation. Supplementation of α-ketoglutarate, a metabolite converted by *isocitrate dehydrogenase alpha-1* (*idha-1*) in the TCA cycle, increases lifespan in *C. elegans*. However, whether *idha-1* can regulate lifespan in *C. elegans* remains unknown. Here, we reported that the expression of *idha-1* modulates lifespan and oxidative stress tolerance in *C. elegans*. Transgenic overexpression of *idha-1* extends lifespan, increases the levels of NADPH/NADP^+^ ratio, and elevates the tolerance to oxidative stress. Conversely, RNAi knockdown of *idha-1* exhibits the opposite effects. In addition, the longevity of *eat-2 (ad1116)* mutant via dietary restriction (DR) was reduced by *idha-1* knockdown, indicating that *idha-1* may play a role in DR-mediated longevity. Furthermore, *idha-1* mediated lifespan may depend on the target of rapamycin (TOR) signaling. Moreover, the phosphorylation levels of S6 kinase (p-S6K) inversely correlate with *idha-1* expression, supporting that the *idha-1*-mediated lifespan regulation may involve the TOR signaling pathway. Together, our data provide new insights into the understanding of *idha-1* new function in lifespan regulation probably via DR and TOR signaling and in oxidative stress tolerance in *C. elegans*.

## 1. Introduction

Aging is an irreversible progression accompanied by physiologic function decline progressively and eventually leads to death [1]. Aging and metabolism are highly intertwined [2]. Several mechanisms have been reported to compose a complex regulation of lifespan, and one of the evolutionarily conserved mechanisms is dietary restriction (DR) [3,4]. DR prolongs the lifespan of yeast, *C. elegans*, *Drosophila*, and mammals [5]. DR not only affects metabolic rate but also promotes longevity and stress resistance via several signaling pathways [5,6].

Multiple signaling pathways are involved in the mechanism of dietary restriction, such as insulin/IGF-1-like signaling, sirtuin, and target of rapamycin (TOR) signaling [7]. Among those pathways, the TOR pathway is closely linked to DR [8,9]. TOR acts as a nutrient-sensing signaling pathway in which its activity increases when food is abundant and vice versa [7,8,10]. TOR controls protein synthesis by phosphorylating ribosomal protein S6 kinase (S6K) and restrains the activity of transcription factor PHA-4 and FOXO/DAF-16, as well as inhibits autophagy and the expression of stress-related genes [7,9]. Repressing TOR activity by rapamycin is sufficient to extend the lifespan in *C. elegans*, *Drosophila*, and mice [11]. Several studies indicated that RNAi silencing TOR or its downstream target S6K can increase lifespan in yeast, *Drosophila*, and *C. elegans* [12,13,14].

Metabolic alternation plays an important role during the aging process [15]. The manipulation of genes and metabolites in the tricarboxylic acid (TCA) cycle has been reported to regulate lifespan [16]. Lifespan extension occurs when adding either α-ketoglutarate, succinate, pyruvate, malate, fumarate, or citrate which are all metabolites generated by the TCA cycle, in *C. elegans* or in *Drosophila* [17,18,19,20,21,22]. Among these metabolites, α-ketoglutarate supplementation not only prolongs the lifespan in worms and flies but also enhances longevity in mice [23] and may also help in promoting human health [24]. Besides metabolites supplementation, the manipulation of the enzymes in the TCA cycle also can modulate lifespan. The inhibition of *gdh-1* and *dld-1*, which both increase the α-ketoglutarate level, can extend the lifespan in *C. elegans* [17,25]. Isocitrate dehydrogenase (IDH) is a rate-limiting enzyme which catalyzes isocitrate to become α-ketoglutarate in the TCA cycle [26]. IDH composes three isoenzymes, IDH1, IDH2, and IDH3, which are conserved in eukaryotes. IDH1 and IDH2 are NADP^+^ dependent, whereas IDH3 is NAD^+^ dependent. In humans, IDH3 is a heterotetramer containing two alpha subunits (IDH3A), one beat subunit (IDH3B) and one gamma subunit (IDH3G). However, the role of isocitrate dehydrogenase in lifespan regulation remains unknown.

Here, we report that expression of *isocitrate dehydrogenase alpha-1* (*idha-1*), the ortholog of human IHD3A, regulates lifespan and modulates oxidative stress tolerance in *C. elegans.* Overexpression of *idha-1* prolongs lifespan, increases oxidative stress resistance, and elevates the levels of the NADPH/NADP^+^ ratio. Oppositely, RNAi knockdown of *idha-1* displays reduced lifespan, lowered tolerance to oxidative stress, and diminished levels of the NADPH/NADP^+^ ratio. Mechanistically, the knockdown of *idha-1* partially abolishes the longevity in *eat-2 (ad1116)* dietary restriction (DR) mutant, suggesting *idha-1* participates in DR-mediated longevity. Furthermore, the knockdown of *idha-1* does not significantly block the enhanced lifespan of the *rsks-1 (ok1255)* mutant, implying the involvement of TOR signaling. Moreover, the phosphorylation levels of S6 kinase (p-S6K) inversely correlate with the *idha-1* expression, supporting that the *idha-1*-mediated lifespan regulation may involve the TOR signaling pathway. Our study sheds new light on the function of *idha-1* in lifespan regulation and in oxidative stress response.

## 2. Results

### 2.1. Isocitrate Dehydrogenase Alpha-1 Expression Regulates Lifespan in C. elegans

Supplementation of α-ketoglutarate was reported to extend the lifespan in *C. elegans.* Hence, we asked whether overexpression of *isocitrate dehydrogenase alpha-1 (idha-1)*, which produces α-ketoglutarate, can enhance lifespan in *C. elegans*. We generated *idha-1* overexpression transgenic worm under its endogenous promoter *Pidha-1::idha-1* (also called *idha-1 o/e*) as well as the control transgenic worm *Pidha-1::GFP* with *GFP* under *idha-1* endogenous promoter. Our data indicated that *Pidha-1::GFP* expresses ubiquitously (Appendix A), in the pharynx, body wall muscle, intestine, and nervous system in *C. elegans* (Appendix A–e). Overexpression of *idha-1* by *Pidha-1::idha-1* exhibited an increase in 24.2% in mean lifespan compared with the control *Pidha-1::GFP* (Figure 1a, Appendix A). RT-QPCR analysis also showed a 1.8-fold increase in *idha-1* mRNA levels in the *idha-1* overexpression worms compared with the control (Figure 1b). Overexpression of *idha-1* displayed a 1.75-fold significant increase in IDHA-1 protein levels compared with that in the control (Figure 1c,d). On the other hand, the knockdown of *idha-1* by RNAi targeting on either 5’ or 3’ of the *idha-1* coding region (named as *idha-1* (RNAi) 5’ and *idha-1* (RNAi) 3’) in wild type N2 significantly decreased 16.4% and 19.4% in mean lifespan, respectively, compared with the control by empty vector (*EV*) (Figure 1e, Appendix A). RT-QPCR analysis indicated that knockdown of *idha-1* by *idha-1* (RNAi) 5’ and *idha-1* (RNAi) 3’ reduced about 51% and 52% in *idha-1* mRNA levels compared with the control (Figure 1f). Knockdown of *idha-1* by *idha-1* (RNAi) 5’ and *idha-1* (RNAi) 3’ both lowered about 68% and 53% in IDHA-1 protein levels compared with the control (Figure 1g,h). Together, the results indicate that expression of *idha-1* can modulate lifespan in *C. elegans.*

### 2.2. Expression of idha-1 Modulates α-Ketoglutarate Levels

Since *idha-1* expression, which produces α-ketoglutarate, can regulate lifespan in *C. elegans*, we asked whether the expression of *idha-1* can modulate α-ketoglutarate levels. Overexpression of *idha-1* by *Pidha-1::idha-1* significantly increased 1.29-fold in α-ketoglutarate levels compared with the control (Figure 2a). Conversely, the knockdown of *idha-1* by *idha-1* (RNAi) 5’ and *idha-1* (RNAi) 3’ reduced α-ketoglutarate levels to about 0.75- and 0.85-fold, respectively, compared with the control (Figure 2b). The data showed that expression of *idha-1,* which regulates lifespan, can also modulate α-ketoglutarate levels.

### 2.3. Expression of idha-1 Manages the Tolerance to Oxidative Stress and Levels of NADPH/NADP^+^ Ratio 

Longevity organism is usually associated with better stress resistance [27,28,29,30]. To examine whether the expression of *idha-1* which regulates lifespan is also associated with the tolerance to oxidative stress, we performed an oxidative stress assay with the *idha-1* overexpression and knockdown worms and their controls under paraquat-induced oxidative stress. The *idha-1* overexpression strain exhibited significantly increased survival under 48 and 72 h of paraquat treatment compared with the control (Figure 3a). On the other hand, the knockdown of *idha-1* in N2 significantly reduced the tolerance to oxidative stress compared with the control at 48 and 72 h of the paraquat treatment (Figure 3b). Thus, these data suggest that the expression of *idha-1* is associated with the tolerance to oxidative stress by paraquat.

NADPH can reduce glutathione disulfide (GSSG) to glutathione (GSH) to lower free radical levels. The ratio of NADPH/NADP^+^ level is considered an index for oxidative stress tolerance. The higher the NADPH level, the better the oxidative stress tolerance [28]. Therefore, we measured the NADPH/NADP^+^ ratio in both *idha-1* overexpression and knockdown worms. Overexpression of *idha-1* displayed a significant increase in NADPH/NADP^+^ ratio compared with that in the control (Figure 3c). On the other side, the knockdown of *idha-1* showed significantly decreased levels of NADPH/NADP^+^ ratio compared with the control (Figure 3d). Together, these results reveal that the expression of *idha-1*, which regulates lifespan, also manages the tolerance to oxidative stress and the levels of NADPH/NADP^+^ ratio in *C. elegans*.

### 2.4. idha-1 Plays A Role in Dietary Restriction Induced Longevity

Reduced fecundity is associated with dietary restriction (DR)-mediated longevity. The long-lived *eat-2 (ad1116)* mutant, which exhibits reduced pharyngeal pumping rate for lower food intake and is used as a DR model, exhibits a reduction in progeny production. The supplementation of α-ketoglutarate was reported to be unable to further prolong the extended lifespan in the *eat-2 (ad1116)* mutant, suggesting the longevity by α-ketoglutarate is similar to that by DR. Therefore, we asked whether the long-lived *idha-1* overexpression transgenic strain which shows an elevated α-ketoglutarate level displays reduced fecundity. We measured the brood size and found a significant decrease in the number of progenies from day 2 to day 6 in the *idha-1* overexpression transgenic line compared with the control (Figure 4a). The average of the total progeny number significantly declines by about 50% in the *idha-1* overexpression line compared with the control (Figure 4b). Interestingly, we found the *idha-1* mRNA levels are elevated in both the DR-treated worms by fasting and the *eat-2 (ad1116)* mutant DR model compared with their controls in the GEO data (Appendix A) as well as in the verification by quantitative RT-PCR analysis (Appendix A). The expression of many other TCA cycle genes is also upregulated in the DR-treated worms by fasting and the *eat-2 (ad1116)* mutant DR model compared with their controls in the GEO data (Appendix A). The data imply the longevity of *idha-1* overexpression may also be involved in DR-induced longevity. Moreover, the knockdown of *idha-1* significantly diminished the extended lifespan in *eat-2 (ad1116)* (Figure 4c, Appendix A), which indicates the longevity in *eat-2 (ad1116)* mutant partially depends on *idha-1*. Overall, the results support our hypothesis that *idha-1* plays a role in dietary restriction-mediated longevity in *C. elegans*.

### 2.5. Expression of idha-1 May Regulate Lifespan via Inversely Modulating TOR Signaling

Dietary restriction is known to interact through multiple signaling pathways, such as the insulin/IGF-1 signaling (IIS) pathway, AMP-activated protein kinase (AMPK) pathway and Target of Rapamycin (TOR) pathway [7]. Lifespan extension caused by reducing the IIS pathway and TOR pathway signaling or activating the AMPK pathway is conserved from yeast to mammals [4]. Our finding suggests that *idha-1* may be involved in DR-induced longevity. To further study the underlying molecular mechanism of *idha-1*-mediated lifespan regulation, we examined the lifespan by reducing *idha-1* expression in the *daf-16 (mu86)*, *aak-2 (gt33)*, and *rsks-1 (ok1255)* mutant strains related to IIS, AMPK, and TOR pathways, respectively. Knockdown of *idha-1* decreased the lifespan in short-lived *daf-16 (mu86)* (Figure 5a, Appendix A), suggesting that *idha-1*-mediated lifespan is independent of *daf-16*. Similarly, the knockdown of *idha-1* also reduced the lifespan in the short-lived *aak-2 (gt33)* mutant (Figure 5b, Appendix A), indicating that *idha-1*-mediated lifespan is independent of *aak-2*. On the other hand, the knockdown of *idha-1* did not significantly shorten the lifespan in long-lived *rsks-1 (ok1255)* mutant (Figure 5c, Appendix A), implying that *idha-1*-mediated lifespan may depend on reducing TOR signaling.

To strengthen our hypothesis that *idha-1*-mediated lifespan may depend on lowered TOR signaling, we examined the levels of p-S6K, which is a downstream effector of TOR signaling in both *idha-1* overexpression and knockdown strains. The long-lived *idha-1* overexpression strain displayed significantly lower p-S6K levels compared with the control (Figure 6a,b), whereas the short-lived *idha-1* knockdown lines showed the opposite results with the elevated p-S6K levels compared with the control (Figure 6c,d). Together, the data support the finding that the *idha-1*-mediated lifespan depends on TOR signaling in *C. elegans.*

## 3. Discussion

Despite as an enzyme in the TCA cycle, in this study, we uncover a new function of *idha-1* in lifespan regulation and oxidative stress response in *C. elegans*. Those phenotypes are associated with the levels of NADPH/DADP+ ratio. This is in accordance with the report that Sirt3 can activate mitochondrial isocitrate dehydrogenase 2 to increase NADPH levels in response to caloric restriction in mice [31]. Our previous study also reported that the longevity and oxidative stress resistance by *ribose-5-phosphate isomerase* knockdown are associated with the levels of NADPH/DADP^+^ ratio in *Drosophila* [28]. In addition, we disclose that the mechanism of lifespan regulation by *idha-1* could be mediated by dietary restriction and TOR signaling. This is consistent with the previous findings that α-ketoglutarate supplementation prolongs lifespan via inhibiting TOR signaling in *C. elegans* and *Drosophila* [17,20]. Moreover, those *idha-1*-mediated phenotypes not only occur in *C. elegans* but also are associated with an allele related to *isocitrate dehydrogenase* in *Drosophila* by selected breeding [32], suggesting that lifespan regulation and the oxidative stress response by *idha-1* may be evolutionary conserved. IDHA-1 is conserved among the *Caenorhabditis* genus (Appendix A).

The TCA cycle is an energy-producing and metabolism process. Recent studies revealed that the supplementation of TCA cycle metabolites, such as α-ketoglutarate, malate, and fumarate, benefits longevity [17,22]. Not only can the supplementation of certain metabolites in the TCA cycle extend lifespan, but also genetic manipulation of some genes in the TCA cycle can regulate longevity. The knockdown of *ogdh-1, sdha-1,* and *dld-1* can increase α-ketoglutarate levels and also extend the lifespan in *C. elegans* [17,22,25]. Apart from manipulating the TCA cycle directly, a previous study showed that long-lived rodents tend to upregulate TCA cycle genes or show no decline in TCA cycle function [33,34]. However, among all these studies, no report yet shows single TCA cycle gene overexpression can prolong lifespan. Here, we provide the first new strategy of lifespan extension in nematode by overexpressing the TCA cycle gene *idha-1* in *C. elegans*.

Better oxidative stress tolerance is usually accompanied by a higher antioxidant capacity. Since the elevation of NADPH levels shows better resistance against ROS and free radicals, the NADPH/NADP^+^ ratio can reflect the tolerance against oxidative stress [28,35]. Here, we found *idha-1* overexpression strain exhibits higher NADPH levels, and the knockdown of *idha-1* reduced NADPH levels and vice versa. Although the supplementation of α-ketoglutarate has been shown to extend lifespan, it is unable to elevate the oxidative stress tolerance across the species [17,20,23]. The discrepancy could be the additional genetic effect of *idha-1* expression, which can modulate NADPH levels. The other possibility is that *idha-1* expression may regulate some antioxidant gene expression. Our study provided another new piece of evidence that *idha-1* expression modulates oxidative stress tolerance in *C. elegans*.

Moreover, we discovered that *idha-1* overexpression may be linked to the DR-induced longevity phenotype and the reduction in brood size [36]. To investigate the direct relationship between *idha-1* and DR, we perform a lifespan assay under DR model conditions and find out whether DR-induced longevity may partially rely on *idha-1* expression. DR-induced longevity includes several nutrient-signaling pathways, such as the IIS pathway, AMPK pathway, and TOR pathway [37]. Unraveling the potential mechanism *idha-1*-mediated lifespan regulation may depend on, we knockdown *idha-1* in the mutants of these pathways. The result points out that only the TOR signaling pathway, but not the IIS or AMPK, is responsible for *idha-1*-mediated lifespan regulation. Together, these results support our hypothesis that *idha-1* plays a role in DR-induced longevity through the TOR signaling pathway.

Besides the IIS, AMPK, and TOR pathways, autophagy and sirtuins are also involved in DR-induced longevity [38]. The relationship between the TOR pathway and autophagy among species is well documented. The inhibition of the TOR signaling pathway may cause the activation of autophagy. To further validate the relation between autophagy and *idha-1* mediation, we may investigate the molecular changes of some autophagy-related genes, such as *bec-1, lgg-1*, and *sqst-1* [39], or even measure the autophagy flux in the future study. 

Aside from autophagy, sirtuins may also involve in *idha-1*-mediated lifespan regulation. Sirtuins are a family of NAD^+^-dependent protein deacetylases, which use NAD^+^ to remove acetyl moieties on histones and proteins [40]. Sirt1 is one of the sirtuins in humans which participates in DR-induced longevity [41]. The activity of Sirt1 accompanies the elevation of NAD^+^ levels during DR. The orthologue of *Sirt1*, *sir-2.1*, can partially regulate DR-induced longevity by activating DAF-16/FOXO [42]. Based on the function of *idha-1*, which converts NAD^+^ to NADH, we assumed overexpression of *idha-1* may elevate the NADH level. However, we did not observe increased NADH levels when overexpressing *idha-1*. Surprisingly, we found an elevation of the NAD^+^ level, which is associated with DR-mediated longevity, upon *idha-1* overexpression (Appendix A). A previous study demonstrated that the elevation of NAD^+^ level accompanies better mitochondrial efficiency and also proposes that the elevation of the NAD^+^ level is a consequence of DR [43]. As our data show that *idha-1* overexpression elevates NAD^+^ levels and participates in DR-induced longevity, it may imply the involvement of *sir-2.1*. We may investigate the role of *sir-2.1* in *idha-1*-mediated lifespan regulation in future work. 

In summary (Figure 7), we disclose the new role of *idha-1* in lifespan regulation and oxidative stress tolerance. Our study demonstrates that *idha-1* regulates lifespan by participating in DR-induced longevity inversely through the TOR signaling pathway and modulates oxidative stress tolerance via the levels of NADPH/NADP^+^ ratio. Increasing amounts of evidence demonstrate that TCA metabolites and enzymes control human physiology and diseases [44,45]. The information from this study will help in the development of α-ketoglutarate as dietary supplementation in promoting human health [24].

## 4. Materials and Methods

### 4.1. Generation of RNA Interference Constructs against idha-1

To generate the RNA interference (RNAi) constructs targeting *idha-1*, we designed two primer sets for two different coding regions of idha-1, one includes exon 2 and exon 3 for 312bp amplified by the primers *idha-1* 5’ forward: 5’-GCCGCCGATGCC-3’ and *idha-1* 5’ reverse: 5’-CTCGTGCTCAATTCCAGAG-3’, and the other includes exon 7 and exon 8 for 267bp amplified by the primers *idha-1* 3’ forward: 5’-GAAGCCTACCTCGACACAG-3’ and *idha-1* 3’ reverse: 5’-CATGTAACGGAGCATCATG-3’. The amplicons were cloned into the *HindIII* site in the L4440 vector, and the resultant constructs were named *idha-1* (RNAi) 5’ and *idha-1* (RNAi) 3’ separately and were transformed into HT115 for the RNAi experiments. The preparation of an RNAi construct containing bacteria for the RNAi experiments was described previously [29,30].

### 4.2. Generation of idha-1 Overexpression Transgenic Worms

To establish the *idha-1* overexpression transgenic construct, we amplified the *idha-1* genomic DNA containing 1 kb upstream promoter region and the whole *idha-1* genomic DNA to be cloned into L3691 vector by using the primers p1-F idha-1-pstI (5’-TTTCTGCAGTCGAAGTTGTCAAAATCCACGAGA-3’) and p2-R-idha-1-kpnI (5’-TTTGGTACCTCTGAACCCTGGAAACAAAAATATTT-3’). The resultant *idha-1* overexpression transgenic construct was named *Pidha-1::idha-1*. For the control transgenic construct, we cloned the 1 kb *idha-1* upstream promoter region into L3691 to express GFP and named the construct *Pidha-1::gfp*. To generate the *idha-1* overexpression transgenic and the control transgenic worms, N2 worms were prepared and synchronized via timed eggs prepared as described previously [30]. The plasmid construct was diluted in ddH_2_O and injected into young adult worms (4 to 8 eggs) with co-injection vector *Psur-5::rfp* at a total concentration of 20 ng/µL. The progenies of the injected worms were then screened for RFP expression and with GFP as well for the control and established as the transgenic lines. Worms expressing RFP were picked and transferred onto new plates every 5 days to maintain a stable line and verified by PCR. The control transgenic worm *Pidha-1::gfp* was used to examine the tissue expression patterns of the GFP driven by the *idha-1* promoter.

### 4.3. Lifespan Analysis and Oxidative Stress Assay

Lifespan assay was performed at 20 °C as described previously [30]. Parent worms were grown to the day-1-adult stage on NGM plates seeded with *E. coli* strain OP50. Nearly 20 adult worms were transferred to RNAi-containing plates or normal NGM plates for 6 h and picked off. The offspring were synchronized on the plates by timed egg lay assay. After the offspring grew to the L4 stage, 30 worms were picked onto each plate, and at least 120 worms existed in each group per trial. Worms were transferred to new plates and the number of deaths was counted every 2 days until all worms died. For oxidative stress assay, 50 worms were picked onto a new plate containing 10 mM paraquat (1, 1’-Dimethyl-4, 4’-bipyridinium dichloride, Sigma-Aldrich). The number of dead worms was calculated every 24 h. The statistical analyses of lifespan and oxidative stress assay were performed by OASIS 2 and *p*-values were calculated by log-rank test for lifespan and *t*-test or one-way ANOVA test for oxidative stress assay.

### 4.4. Western Blot

Worms with or without specific treatment were collected and lysed by whole-cell extract (WCE) buffer containing 20 mM HEPES (pH 7.5), 75 mM NaCl, 2.5 mM MgCl_2_, 0.1 mM EDTA, 0.5% Triton X-100, 0.1 mM Na_3_VO_4,_ 50 mM NaF and protease inhibitor (Roche, Cat#04 693 124 001). The samples with an equal amount of protein were prepared as described previously for the western blot [30]. The primary antibodies used were listed below: IDHA-3A (Genetex, GTX114486), beta-actin (GeneTex, GTX109639), and p-S6K (Cell Signaling, Cat# 9209). The secondary antibody is goat anti-rabbit IgG HRP-conjugate antibodies (JIR, Cat#111-035-003). The resultant NC membrane was then incubated with ECL (Millipore, Cat#WBKLS0500), and detected by ImageQuant (GE Healthcare, LAS 4000 mini). Statistical analysis was performed by ImageJ and *p*-values were calculated by *t*-test or one-way ANOVA test.

### 4.5. RNA Extraction and qRT-PCR

The worms were washed with an M9 buffer three times and collected in a 1.5-mL micro-centrifuge tube. The RNA extraction and RT-QPCR procedures were described previously [30]. The RNA pellet was dissolved by 10 μL DEPC-treated ddH_2_O and measured the concentration by NanoDrop 2000 (Thermo). The qRT-PCR was performed by using the Step One Plus Real-Time PCR system (ABI). The relative expression levels of genes between different samples were calculated on the basis of ΔΔCt (threshold cycles) values and *act-1* was used as the internal control for normalization. Statistical analysis was calculated by *t*-test or one-way ANOVA test. The primers for *idha-1* are the forward primer 5’-GCACGCGAACAAAGTTGGAC-3’ and the reverse primer 5’-CTTCAAGGGAACGGCATGGA-3’. The primers for *act-1* are the forward primer 5’-CTCTTGCCCCATCAACCATGA-3’ and the reverse primer 5’-TTGCGGTGAACGATGGATGG-3’.

### 4.6. Measurement of α-Ketoglutarate Level

The α-ketoglutarate levels were measured by a colorimetric quantification kit (BioVision, Cat# K677-100) for Figure 3c and by a bioluminescent assay-based kit (NADP/NADPH-Glo™ assay; Promega, Cat#G9081) for Figure 3d according to the manufacturer’s protocols. Worms were collected into 1.5-mL centrifuge tubes and homogenized in the extraction buffer provided with the kit. The samples were centrifuged at 13,000 rpm for 5 min and the supernatants were transferred into new labeled tubes. To detect the α-ketoglutarate level, 50 μL samples were transferred into a 96-well ELISA plate. 2 μL of cycling enzyme mix and 48 μL of buffer mix were added in each well and incubated at 37 °C for 30 min. The plate was read under OD 570 nm with an ELISA reader, and the α-ketoglutarate level was calculated according to the standard curve and normalized by protein concentration.

### 4.7. NADPH/NADP^+^ and NAD^+^/NADH Quantification

The NADPH/NADP^+^ and NAD^+^/NADH quantifications were measured by using the quantification colorimetric kits (BioVision, Cat#: K337-100 and K347-100). The worms were collected into a 1.5-mL eppendorf tube and homogenized by the extraction buffer. 50 μL of each sample was transferred into a 96-well ELISA plate according to the manufacturer’s protocols. Calculated the ratio according to the standard curve.

## Figures and Tables

**Figure 1 ijms-24-00612-f001:**
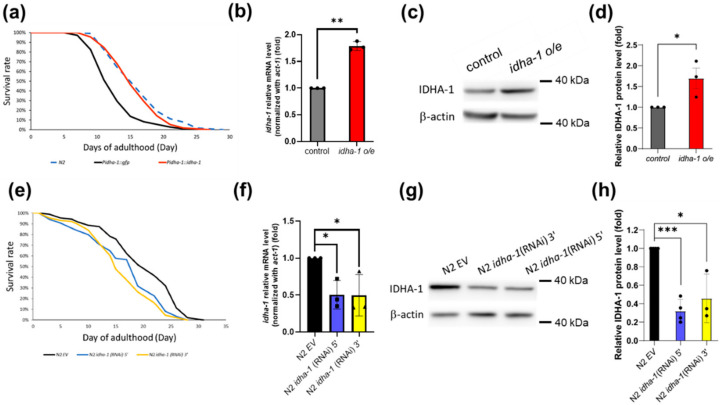
Expression of *idha-1* regulates lifespan in *C. elegans.* (**a**) *idha-1* overexpression (red line, *Pidha-1::idha-1*, mean lifespan: 15.9 days, n = 112) exhibits a 24.2% increase in mean lifespan extension (*p* < 0.0001, log-rank test) compared with the control (black line, *Pidha-1::gfp,* mean lifespan: 12.8 days, n = 109). N2 (blue dotted line, mean lifespan: 16.4 days, n = 115). (**b**) The *idha-1* overexpression strain (red bar) showed a nearly 1.8-fold increase in *idha-1* mRNA levels compared with the control (grey bar) normalized with *act-1*. (** *p* < 0.01, *t*-test). (**c**,**d**) The elevated IDHA-1 protein levels were detected in *idha-1* overexpression worms (*idha-1 o/e*). (* *p* < 0.05, *t*-test). (**e**) Knockdown of *idha-1* (blue line, N2 *idha-1* (RNAi) 5’, mean lifespan: 16.8 days, n = 173; and yellow line, N2 *idha-1* (RNAi) 3’, mean lifespan: 16.2 days, n = 167) reduces mean lifespan (−16.4% and −19.4% respectively, both *p* < 0.0001, log-rank test) compared with the control (black line, N2 EV, mean lifespan: 20.1 days, n = 169). (**f**) The *idha-1* knockdown strains (blue bar, N2 *idha-1* (RNAi) 5’ and yellow bar, N2 *idha-1* (RNAi) 3’) both decreased 0.5-fold in *idha-1* mRNA levels compared with the control (black bar, N2 EV) normalized with *act-1*. (* *p* < 0.05, one-way ANOVA test). (**g**,**h**) The decreased IDHA-1 protein levels were detected by the knockdown of *idha-1* (*** *p* < 0.001 and * *p* < 0.05 respectively, one-way ANOVA test). Each bar represents mean ± SD.

**Figure 2 ijms-24-00612-f002:**
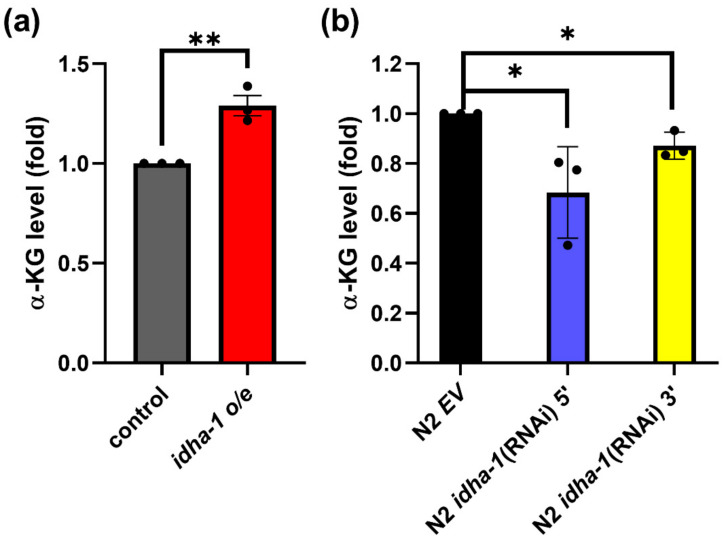
Expression of *idha-1* mediates α-ketoglutarate levels. (**a**) Overexpression of *idha-1* (red bar) significantly increases α-ketoglutarate levels compared with the control (grey bar). (** *p* < 0.01, *t*-test). (**b**) Knockdown of *idha-1* (blue bar, N2 *idha-1* (RNAi) 5’ and yellow bar, N2 *idha-1* (RNAi) 3’) shows lowered α-ketoglutarate levels compared with the control (black bar, N2 EV). (* *p* < 0.05, one-way ANOVA test). Each bar represents mean ± SD, n = 3.

**Figure 3 ijms-24-00612-f003:**
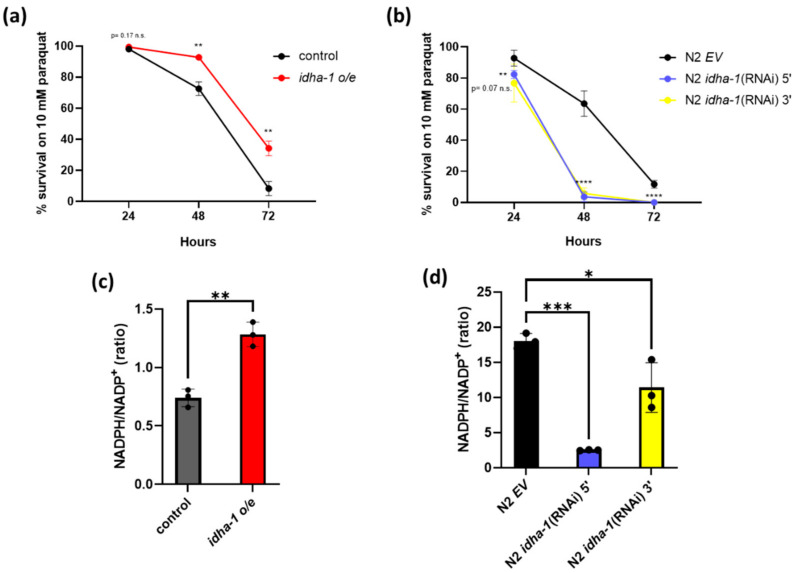
Expression of *idha-1* modulates the tolerance to oxidative stress and NADPH/NADP^+^ levels. (**a**) Overexpression of *idha-1* (red dot, n = 263) significantly increases oxidative stress tolerance after 48 and 72 h of paraquat treatment (** *p*< 0.01, *t*-test) compared with the control (black dot, n = 252). (**b**) Knockdown of *idha-1* (blue dot, N2 *idha-1* (RNAi) 5’, n = 164; and yellow dot, N2 *idha-1* (RNAi) 3’, n = 158) significantly reduces the tolerance to oxidative stress compared with the control (black dot, N2 EV, n = 157) after 24-, 48- and 72-h treatment (** *p* < 0.01, **** *p* < 0.0001, one-way ANOVA test). Each dot represents mean ± SD, n = 3. (**c**) Overexpression of *idha-1* (red bar) significantly increases the NADPH/NADP^+^ ratio levels compared with the control (grey bar). (** *p* < 0.01, *t*-test). (**d**) Knockdown of *idha-1* (blue bar, N2 *idha-1* (RNAi) 5’ and yellow bar, N2 *idha-1* (RNAi) 3’) significantly decreases the NADPH/NADP^+^ ratio levels compared with the control (black bar, N2 EV). (*** *p* < 0.001 and * *p* < 0.05, one-way ANOVA test). Each bar represents mean ± SD, n = 3.

**Figure 4 ijms-24-00612-f004:**
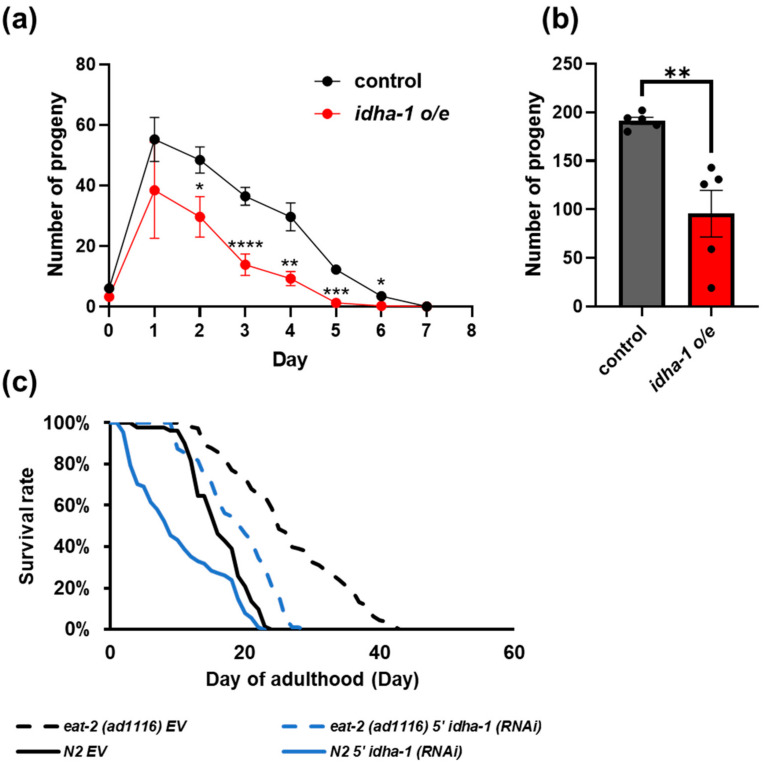
*idha-1* plays a role in dietary restriction-induced longevity. (**a**) Overexpression of *idha-1* (red dot) significantly decreases progeny number between day 2 to day 6 compared with the control (black dot). Each dot plot represents mean ± SD, n = 3 (* *p* < 0.05, ** *p* < 0.01, *** *p* < 0.001, **** *p* < 0.0001, *t*-test) (**b**) Total brood size is significantly diminished in *idha-1* overexpression strain (red bar) compared with the control worm (grey bar). Each dot plot represents mean ± SD, n = 5 (* *p* < 0.05, ** *p* < 0.01, *** *p* < 0.001, **** *p* < 0.0001, *t*-test). (**c**) Knockdown of *idha-1* in *eat-2 (ad1116)* longevity mutant (blue dotted line, mean lifespan: 20.8 days, n = 120) partially abolishes the extended lifespan in *eat-2 (ad1116)* longevity mutant (black dotted line, mean lifespan: 25.7 days, n = 113) compared with the control (black solid line, mean lifespan: 20.1 days, n = 169). Knockdown of *idha-1* in N2 (blue solid line, mean lifespan: 16.8 days, n = 193) displays a reduced lifespan compared with the control (black solid line). (*p* < 0.0001, log-rank test).

**Figure 5 ijms-24-00612-f005:**
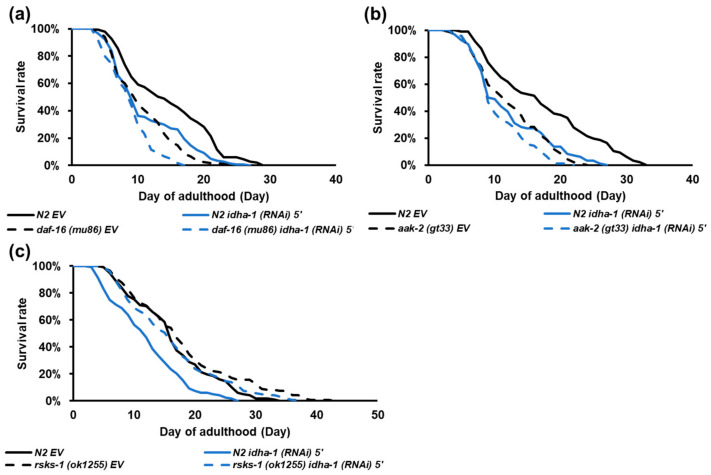
The *idha-1*-mediated lifespan may depend on TOR signaling but not on insulin or AMPK signaling pathway. (**a**) Knockdown of *idha-1* in N2 (blue solid line, mean lifespan: 11.3 days, n = 141) (*p* < 0.0001, log-rank test) shortened lifespan compared with the control (black solid line, mean lifespan: 14.7 days, n = 139). The shortened lifespan caused by *daf-16* mutation *(mu86)* (black dotted line, mean lifespan: 10.9 days, n = 121) (*p* < 0.0001, log-rank test) can be further reduced by *idha-1* knockdown (blue dotted line, mean lifespan: 9.0 days, n = 120) (*p* < 0.0001, log-rank test). (**b**) The shortened lifespan caused by *aak-2* mutation *(gt33)* (black dotted line, mean lifespan: 12.2 days, n = 141) (*p* < 0.0001, log-rank test) can be further decreased by *idha-1* knockdown (blue dotted line, mean lifespan: 10.8 days, n = 117) (*p* < 0.001, log-rank test). Knockdown of *idha-1* in N2 (blue solid line, mean lifespan: 12.2 days, n = 141) (*p* < 0.0001, log-rank test) shortened lifespan compared with the control (black solid line, mean lifespan: 17.2 days, n = 139). (**c**) Knockdown of *idha-1* in long-lived *rsks-1* mutant strain *(ok1255)* (blue dotted line, mean lifespan: 16.3 days, n = 122) (*p* = 0.052, log-rank test) cannot significantly abolish the extended lifespan caused by *rsks-1* mutation (black dotted line, mean lifespan: 18.1 days, n = 122) (*p* < 0.05, log-rank test). Knockdown of *idha-1* in N2 (blue solid line, mean lifespan: 12.1 days, n = 119) (*p* < 0.0001, log-rank test) significantly shortened lifespan compared with the control (black solid line, mean lifespan: 16.4 days, n = 118).

**Figure 6 ijms-24-00612-f006:**
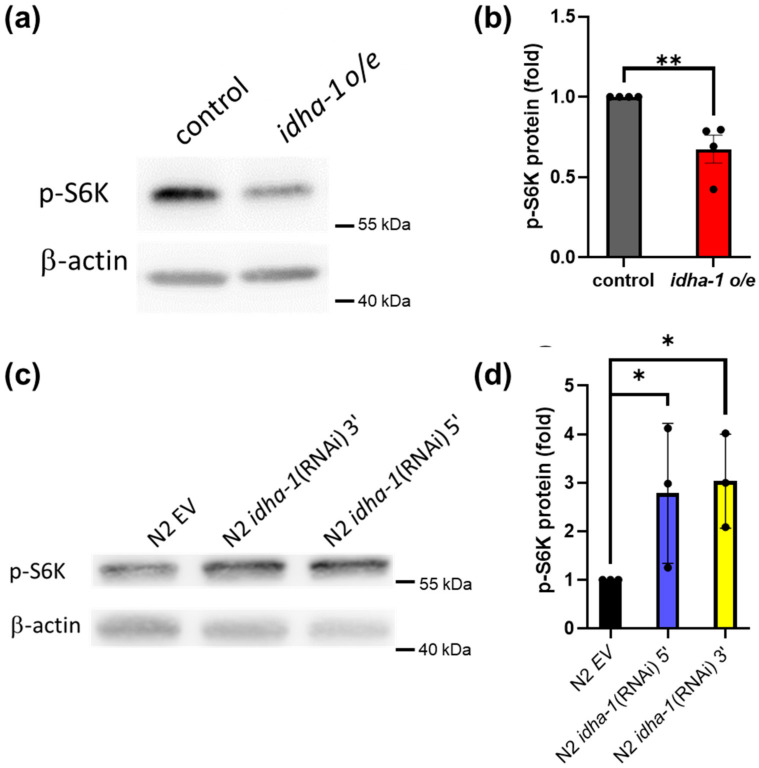
Expression of *idha-1* negatively modulates the phosphorylated S6K levels. The level of phosphorylated S6K (p-S6K), a TOR activity indicator, in each strain was determined and normalized with β-actin level. (**a**,**b**) Long-lived *idha-1* overexpression strain (red bar, *idha-1 o/e*) displays significantly reduced p-S6K levels compared with the control (grey bar) (** *p* < 0.01, *t* test). Each bar represents mean ± SD, n = 4. (**c**,**d**) Short-lived *idha-1* knockdown strains (blue bar, N2 *idha-1* (RNAi) 5’; and yellow bar, N2 *idha-1* (RNAi) 3’) exhibit significantly increased p-S6K levels compared with the control (black bar, N2 EV) (* *p* < 0.05, *t* test). Each bar represents mean ± SD, n = 3.

**Figure 7 ijms-24-00612-f007:**
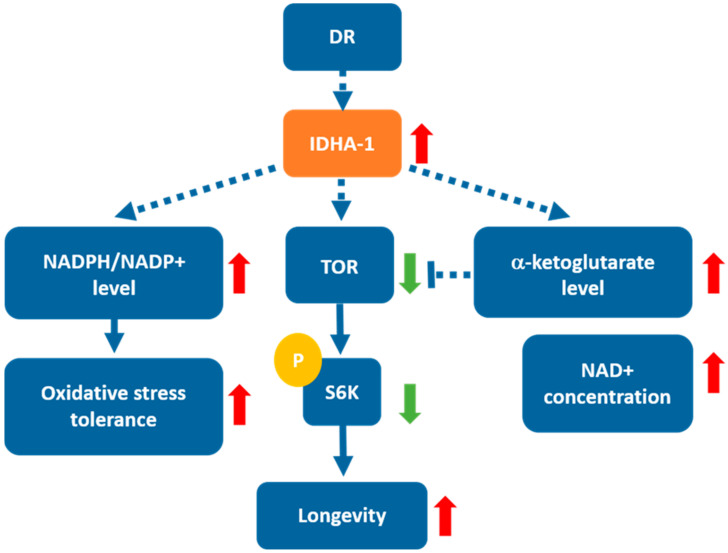
The mechanism of *idha-1*-mediated lifespan regulation and oxidative stress response in *C. elegans*. DR may induce *idha-1* upregulation. Overexpression of *idha-1* results in increased levels of α-ketoglutarate and NAD^+^. In addition, *idha-1* overexpression also leads to elevated levels of NADPH/NADH^+^ ratio which increases oxidative stress tolerance. Moreover, *idha-1* overexpression may negatively regulate TOR signaling to extend lifespan in *C. elegans*.

## Data Availability

Not applicable.

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
