# Peer review of "Isocitrate Dehydrogenase Alpha-1 Modulates Lifespan and Oxidative Stress Tolerance in Caenorhabditis elegans"

_ijms, 2022, doi:10.3390/ijms24010612_

Round 1
Reviewer 1 Report
This paper described that the effects of changes in isocitrate dehydrogenase α-1 expression on the longevity and oxidative stress tolerance of C. elegans. The experiment is well done and the authors present very interesting data. Therefore, I think that it is a suitable work to be published in IJMS. Some corrections are, however, required and the authors should also discuss the comments listed below.
Figure 3, The ratio of NADPH/MADP+ in the control and N2 EV worms is big different. What is the reason for this? Does RNAi have any effect on the ratio of NADPH/MADP+?
The authors described that idha-1 plays a role in dietary restriction mediated longevity in C. elegans. In fact, Idha-1 transcript levels are elevated in the DR-treated worms or eat-2 mutants (Figure S2). I believed that these data are useful information to support the above hypothesis. By the way, it would be interesting to know what the α-KD level is in DR-treated worms and eat-2 mutants. I think that this information strongly supports your hypothesis. I highly recommend measuring the α-KD level.
Since the authors have RNA-seq data in the DR-treated worms and eat-2 mutants, why not also show the expression of other TCA cycle enzyme groups?
Author Response
Response to Reviewer 1:
Comments and Suggestions for Authors
This paper described that the effects of changes in isocitrate dehydrogenase α-1 expression on the longevity and oxidative stress tolerance of C. elegans. The experiment is well done and the authors present very interesting data. Therefore, I think that it is a suitable work to be published in IJMS. Some corrections are, however, required and the authors should also discuss the comments listed below.
Figure 3, The ratio of NADPH/MADP+ in the control and N2 EV worms is big different. What is the reason for this? Does RNAi have any effect on the ratio of NADPH/NADP+?
Answer: We would like to thank the comment from Reviewer 1. The difference of Y-scale in Figure 3C and 3D is because by the different NADPH measurement assay kits. In figure 3C we used colorimetric assay to measure the NADPH/NADP+ ratio. However, colorimetric assay is less sensitive to detect the differences in the experimental groups of Figure 3D. Therefore, we used a bioluminescent assay based kit (NADP/NADPH-Glo™ assay; Promega, Cat#G9081) which has a higher sensitivity to detect NADPH/NADP+ ratio in Figure 3D. We have updated the NADP/NADPH-Glo™ assay information into the Materials and Methods.
Regarding whether RNAi has any effect on the NADPH/NADP+ ratio, we believe RNAi itself should not affect the results of the NADPH/NADP+ ratio. The reduced levels of NADPH/NADP+ ratio by idha-1 RNAi knockdown should derive from the reduced idha-1 expression. Because in our other manuscript currently under review in Antioxidants journal where we showed that RNAi knockdown of ribose-5-phosphate isomerase A-1 (rpia-1) increases the levels of NADPH/NADP+ ratio and enhances the tolerance to oxidative stress in C. elegans. Similarly, in our 2012 Aging Cell paper by Wang, C.T. et al. ([1], doi: 10.1111/j.1474-9726.2011.00762.x. ), we also showed that RNAi knockdown of Drosophila ribose-5-phosphate isomerase can increase the levels of NADPH/NADP+ ratio. Those may exclude the possibility of RNAi effect itself on the reduced levels of NADPH/NADP+ ratio in Figure 3D.
The authors described that idha-1 plays a role in dietary restriction mediated longevity in C. elegans. In fact, Idha-1 transcript levels are elevated in the DR-treated worms or eat-2 mutants (Figure S2). I believed that these data are useful information to support the above hypothesis. By the way, it would be interesting to know what the α-KD level is in DR-treated worms and eat-2 mutants. I think that this information strongly supports your hypothesis. I highly recommend measuring the α-KD level.
Answer: We would like to thank Reviewer 1 for the insight suggestion. Indeed, in the 2014 Nature paper by Chin et al.- “The metabolite α-ketoglutarate extends lifespan by inhibiting ATP synthase and TOR” [2], they have shown that starvation, which is a brief form of dietary restriction, can elevate α-ketoglutarate levels in C. elegans in their Figure 4d as shown below. In addition, the authors also described as quoted below that “α-ketoglutarate levels are increased in starved yeast and bacteria (Brauer, M.J., et al., 2006, PNAS, v103, p19302, [3]), in the liver of starved pigeons (Kaminsky, Y.G., et al., 1982, Comp. Biochem. Physio. B, v73, p957, [4]), and in humans after physical exercise (Brugnara, L. et al., 2012, PLoS ONE, v7, e40600, [5])”. The information above may support the hypothesis that α-ketoglutarate levels can be increased in DR-treated organisms.
Since the authors have RNA-seq data in the DR-treated worms and eat-2 mutants, why not also show the expression of other TCA cycle enzyme groups?
Answer: We would like to thank the constructive suggestion from Reviewer 1. We have analyzed the expression levels of the other TCA cycle genes in the RNA-seq data from the DR-treated worms and eat-2 mutant as well as their controls, and added the data into Figure S2 (d) and (e) and also shown below and mentioned it in the text.
References:
- Wang, C. T.; Chen, Y. C.; Wang, Y. Y.; Huang, M. H.; Yen, T. L.; Li, H.; Liang, C. J.; Sang, T. K.; Ciou, S. C.; Yuh, C. H.; Wang, C. Y.; Brummel, T. J.; Wang, H. D., Reduced neuronal expression of ribose-5-phosphate isomerase enhances tolerance to oxidative stress, extends lifespan, and attenuates polyglutamine toxicity in Drosophila. Aging Cell 2012, 11, (1), 93-103.
- Chin, R. M.; Fu, X.; Pai, M. Y.; Vergnes, L.; Hwang, H.; Deng, G.; Diep, S.; Lomenick, B.; Meli, V. S.; Monsalve, G. C.; Hu, E.; Whelan, S. A.; Wang, J. X.; Jung, G.; Solis, G. M.; Fazlollahi, F.; Kaweeteerawat, C.; Quach, A.; Nili, M.; Krall, A. S.; Godwin, H. A.; Chang, H. R.; Faull, K. F.; Guo, F.; Jiang, M.; Trauger, S. A.; Saghatelian, A.; Braas, D.; Christofk, H. R.; Clarke, C. F.; Teitell, M. A.; Petrascheck, M.; Reue, K.; Jung, M. E.; Frand, A. R.; Huang, J., The metabolite α-ketoglutarate extends lifespan by inhibiting ATP synthase and TOR. Nature 2014, 510, (7505), 397-401.
- Brauer, M. J.; Yuan, J.; Bennett, B. D.; Lu, W.; Kimball, E.; Botstein, D.; Rabinowitz, J. D., Conservation of the metabolomic response to starvation across two divergent microbes. Proc Natl Acad Sci U S A 2006, 103, (51), 19302-7.
- Kaminsky, Y. G.; Kosenko, E. A.; Kondrashova, M. N., Metabolites of citric acid cycle, carbohydrate and phosphorus metabolism, and related reactions, redox and phosphorylating states of hepatic tissue, liver mitochondria and cytosol of the pigeon, under normal feeding and natural nocturnal fasting conditions. Comp Biochem Physiol B 1982, 73, (4), 957-63.
- Brugnara, L.; Vinaixa, M.; Murillo, S.; Samino, S.; Rodriguez, M. A.; Beltran, A.; Lerin, C.; Davison, G.; Correig, X.; Novials, A., Metabolomics approach for analyzing the effects of exercise in subjects with type 1 diabetes mellitus. PLoS One 2012, 7, (7), e40600.

Reviewer 2 Report
General comments
In this paper, the authors focused on the role of Isocitrate dehydrogenase as a regulator of the lifespan in C. elegans. They show that idha regulates lifespan by participating in DR-induced longevity inversely through TOR signaling pathway and modulates oxidative stress tolerance via the levels of NADPH/NADP +ratio. Overall, the article is nicely written with descriptive information, however several concerns must be responded to improve the manuscript.
1. In the Introduction, the information with respect to Isocitrate dehydrogenase in C. elegans is too brief and may be difficult for readers who are not familiar with idha-1 to understand the rest of the article. Even though C. elegans possess 2 isoforms of the enzyme (idha-1 and 2), the authors never make mention the relevance of this isoform and less, analyze whether this isoform could play any role in the effects that they analyze.
2. In figure 3, the authors have different levels of NADPH/NADP+ ratio (Fig 3c and d), Why is the difference?
3. In figure 4, the authors measure number of progeny, where the maximums peak is at day 1 and only have 60 of eggs. However, it is well known that the progeny production peak is reached at day 2-3 with around 120-150 eggs, just as shown in figure 4b, why is this difference?
4. I suggest the homogenization of survival plots, just as in figure 4C, because figures 1a and e are blurred.
Author Response
Response to Reviewer 2:
Open Review
Comments and Suggestions for Authors
General comments
In this paper, the authors focused on the role of Isocitrate dehydrogenase as a regulator of the lifespan in C. elegans. They show that idha regulates lifespan by participating in DR-induced longevity inversely through TOR signaling pathway and modulates oxidative stress tolerance via the levels of NADPH/NADP+ ratio. Overall, the article is nicely written with descriptive information, however several concerns must be responded to improve the manuscript.
- In the Introduction, the information with respect to Isocitrate dehydrogenase in C. elegans is too brief and may be difficult for readers who are not familiar with idha-1 to understand the rest of the article. Even though C. elegans possess 2 isoforms of the enzyme (idha-1 and 2), the authors never make mention the relevance of this isoform and less, analyze whether this isoform could play any role in the effects that they analyze.
Answer: We would like to thank Reviewer 2 for the suggestion. We have added some more information of Isocitrate dehydrogenase in the Introduction.
- In figure 3, the authors have different levels of NADPH/NADP+ ratio (Fig 3c and d), Why is the difference?
Answer: We would like to thank the comment from reviewer 2. The difference of Y-scale in Figure 3C and 3D is because by the different NADPH measurement assay kits. In figure 3C we used colorimetric assay to measure the NADPH/NADP+ ratio. However, colorimetric assay is less sensitive to detect the differences in the experimental groups of Figure 3D. Therefore, we used a bioluminescent assay based kit (NADP/NADPH-Glo™ assay; Promega, Cat#G9081) which has a higher sensitivity to detect NADPH/NADP+ ratio in Figure 3D. We have updated the NADP/NADPH-Glo™ assay information into the Materials and Methods.
- In figure 4, the authors measure number of progeny, where the maximums peak is at day 1 and only have 60 of eggs. However, it is well known that the progeny production peak is reached at day 2-3 with around 120-150 eggs, just as shown in figure 4b, why is this difference?
Answer: We thank Reviewer 2 for the comment. We used the day 0 to indicate the first day we put the worms in the plate, our day 0 may be the day 1 described in other papers, and our day 1 and day 2 that have the peaks may be the day 2 and day 3 in other papers as Reviewer 2 mentioned and like the data listed below. The reason why the peaks have less eggs than the normal egg number by wild type worms could be they are from the transgenic worms with the co-injection marker Psur-5::rfp and the control Pidha-1::gfp (as the control) or the idha-1 overexpression Pidha-1::idha-1 transgene (as the idha-1 o/e). For example, according to the data as shown below from 2019 microPublication Biology by Madhu B. et al. (doi: 10.17912/2r51-b476, [1]), the egg number in wild type N2 with OP50 can reach 75 and 100 with peaks on day 2 and day 3 respectively, but the dbl-1(nk3) mutant with OP50 showed less egg number with 40 and 45 on day 2 and day 3 respectively. The egg numbers on their day 1 are similar to ours on day 0. Importantly, we found that the idha-1 overexpression transgenic worms produce less eggs than the control transgenic worms under the same genetic background. Why the total number of egg production in the transgenic worms is less than the wild type will be another interesting issue to study.
The data from 2019 microPublication Biology by Madhu B. et al. (doi: 10.17912/2r51-b476 ).
Our original data in the other format for the Figure 4(a):
- I suggest the homogenization of survival plots, just as in figure 4C, because figures 1a and e are blurred.
Answer: Thanks for the constructive comment from Reviewer 2. Accordingly, we have replaced the plots with better resolution for Figure 1a and 1e..
Reference:
- Madhu, B.; Salazar, A.; Gumienny, T., Caenorhabditis elegans egg-laying and brood-size changes upon exposure to Serratia marcescens and Staphylococcus epidermidis are independent of DBL-1 signaling. MicroPubl Biol 2019, 2019.

Reviewer 3 Report
In this research article, the authors provide evidence for the role of idha-1 in regulating lifespan in C. elegans. The authors demonstrate that over-expression of idha- 1 extends the lifespan and enhances oxidative stress tolerance in C. elegans. The authors further provide evidence for the role of idha-1 in eat-2-mediated dietary restriction (DR) and the target of rapamycin (TOR) signaling in longevity.
Fig. 1A). For a better comparison, authors should include N2 control.
The authors suggest that idha-1 is conserved from worms to fly to mice. They should also comment on the conservation of idha-1 in nematodes.
A phylogenetic tree for idha-1 sequence conservation from other Caenorhabditis species will highlight useful.
The authors claim that idha-1 regulates lifespan by modulating oxidative stress tolerance via the NADPH/NADP+ ratio levels. However, this claim is not substantiated by experimental evidence. Although the NADPH and α-ketoglutarate levels are changed in o/e and RNAi animals, these results are insufficient to make the abovementioned claim. A supplementation experiment can be performed to confirm these predictions.
Minor:
Fig 1a/e) Change survival rate to % survival; d/e) relative protein levels
In a table, please provide the exact number of animals (scored or censored) used in each lifespan assay.
Fig S1
The idha-1-reporter expression images appear over-saturated; authors should provide unsaturated images.
For the neuronal images, it needs to be clarified which region of the animal is depicted. In addition, the general identity of Neurons should be mentioned.
Author Response
Response to Reviewer 3:
Open Review
Comments and Suggestions for Authors
In this research article, the authors provide evidence for the role of idha-1 in regulating lifespan in C. elegans. The authors demonstrate that over-expression of idha- 1 extends the lifespan and enhances oxidative stress tolerance in C. elegans. The authors further provide evidence for the role of idha-1 in eat-2-mediated dietary restriction (DR) and the target of rapamycin (TOR) signaling in longevity.
Fig. 1A). For a better comparison, authors should include N2 control.
Answer: We would like to thank Reviewer 3 for the comment. We are sorry that we do not have the N2 control in this same experiment in Figure 1 (a). However, we do have N2 in our very beginning experiment as shown below. We found that the transgenic worm with the co-injection marker only (Psur-5::rfp) in N2 background (yellow solid line, mean= 12.2 days, n=109) showed reduced lifespan compared with the wild type N2 (blue dotted line, mean= 16.4 days, n=115); the control transgenic line with Pidha-1::gfp and the co-injection marker Psur-5::rfp (grey solid line, mean= 12.8 days, n=109) showed no difference compared with the lifespan in the transgenic line with the co-injection marker Psur-5::rfp in N2 background (yellow solid line, mean= 12.2 days, n=109). The idha-1 overexpression transgenic worm with Pidha-1::idha-1 and the co-injection marker Psur-5::rfp (red solid line, mean= 15.9 days, n=112) displayed enhanced mean lifespan with an increase of 24.2% compared to the control transgenic worm (grey solid line, mean= 12.8 days, n=109). As we would like to compare the effect of idha-1 overexpression on the lifespan changes in the same genetic background, therefore we only use the idha-1 overexpression transgenic worm with Pidha-1::idha-1 and the co-injection marker Psur-5::rfp and the control transgenic worm with Pidha-1::gfp and the co-injection marker Psur-5::rfp in the later lifespan assay. Since N2 wild type worm does not contain the transgenic construct (Pidha-1::gfp or Pidha-1::idha-1) and the co-injection marker (Psur-5::rfp), therefore we did not include N2 in our later lifespan experiment as shown in Figure 1 (a).
The authors suggest that idha-1 is conserved from worms to fly to mice. They should also comment on the conservation of idha-1 in nematodes. A phylogenetic tree for idha-1 sequence conservation from other Caenorhabditis species will highlight useful.
Answer: We would like to thank the suggestion from Reviewer 3. We have aligned IDHA-1 protein sequences retrieved from NCBI and by ClustalW using MEGA11 (Molecular Evolutionary Genetics Analysis version 11; Tamura, Stecher, and Kumar 2021). This protein alignment image was taken by SnapGene software (www.snapgene.com) as shown below.
(a)
(b)
Figure S3. IDHA-1 is conserved among the Caenorhabditis genus. (a) The aligned IDHA-1 protein sequences were retrieved from NCBI and aligned by ClustalW using MEGA11 (Molecular Evolutionary Genetics Analysis version 11; Tamura, Stecher, and Kumar 2021). The alignment image was taken by SnapGene software (www.snapgene.com). (b) The unrooted phylogram of IDHA-1 within Caenorhabditis was generated by maximum likelihood method (1024 replicates performed) using MEGA11. The scale bar indicates the substitutions per sites of the horizontal branch lengths; bootstrap values are displayed at the branch nodes in percentage (%).
The authors claim that idha-1 regulates lifespan by modulating oxidative stress tolerance via the NADPH/NADP+ ratio levels. However, this claim is not substantiated by experimental evidence. Although the NADPH and α-ketoglutarate levels are changed in o/e and RNAi animals, these results are insufficient to make the abovementioned claim. A supplementation experiment can be performed to confirm these predictions.
Answer: We would like to thank the comment from Reviewer 3. Better oxidative stress tolerance is usually associated with enhanced longevity. Therefore, we also looked into whether idha-1 expression can modulate oxidative stress tolerance as well as the levels of NADPH/NADP+ ratio and found the positive correlation. However, we did not claim the idha-1-mediated lifespan changes derive from oxidative stress tolerance via the NADPH/NADP+ ratio levels. The lifespan changes and the oxidative stress tolerance by idha-1 are two independent lines as shown in Figure 7. In addition, the supplementation of NADPH may not be easy since it may be easily oxidized while preparation in worm food. Regarding the supplementation of α-ketoglutarate, it has been well documented in the 2014 Nature paper by by Chin et al.- “The metabolite α-ketoglutarate extends lifespan by inhibiting ATP synthase and TOR” [1]. We are sorry for the misunderstanding probably by our writing in our last sentence of the abstract. We have edited that sentence to make it much clear.
Minor:
Fig 1a/e) Change survival rate to % survival; d/e) relative protein levels.
Answer: We thank Reviewer 3 for the constructive suggestion. We have made the corrections for Fig 1d and 1e accordingly. Regarding Fig 1a and 1e, the Y-scales are shown as % already as well as those in Fig 4c and in Fig 5. Therefore, survival rate may be a better indication.
In a table, please provide the exact number of animals (scored or censored) used in each lifespan assay.
Answer: We thank the suggestion and provide the table as below and add as Table S1.
Fig S1, The idha-1-reporter expression images appear over-saturated; authors should provide unsaturated images.
Answer: We truly thank the constructive suggestion from the reviewer 3. We have replaced the images with the unsaturated ones.
For the neuronal images, it needs to be clarified which region of the animal is depicted. In addition, the general identity of Neurons should be mentioned.
Answer: We would like to thank the good point raised from the reviewer 3. By the images, it is hard to clarify which specific neurons they are since they are verified by using specific neuron cell-type reporter to confirm the cell-type by co-localization. After recently consulting with the local C. elegans experts, Dr. Chun-Liang Pan at National Taiwan University and Dr. Bi-Tzen Juang at National Yang Ming Chiao Tung University for their suggestions, we would like to change our indication as neuron to become hyp 7 cells in pharynx region in Fig S1C and seam cells in anterior part in Fig S1D and seam cells in posterior part in Fig S1E.
Reference:
- Chin, R. M.; Fu, X.; Pai, M. Y.; Vergnes, L.; Hwang, H.; Deng, G.; Diep, S.; Lomenick, B.; Meli, V. S.; Monsalve, G. C.; Hu, E.; Whelan, S. A.; Wang, J. X.; Jung, G.; Solis, G. M.; Fazlollahi, F.; Kaweeteerawat, C.; Quach, A.; Nili, M.; Krall, A. S.; Godwin, H. A.; Chang, H. R.; Faull, K. F.; Guo, F.; Jiang, M.; Trauger, S. A.; Saghatelian, A.; Braas, D.; Christofk, H. R.; Clarke, C. F.; Teitell, M. A.; Petrascheck, M.; Reue, K.; Jung, M. E.; Frand, A. R.; Huang, J., The metabolite α-ketoglutarate extends lifespan by inhibiting ATP synthase and TOR. Nature 2014, 510, (7505), 397-401.

Round 2
Reviewer 2 Report
NONE
Author Response
We appreciate Reviewer 2 for all the constructive comments and suggestions in the manuscript revision to substantially improve the quality.

Reviewer 3 Report
The authors have incorporated most of the suggestion that has improved the manuscript. However, although the authors have provided a satisfactory explanation for the exclusion of N2 control from Fig. 1A). For a better comparison, authors must include N2 control data in figure 1A. The figure presented in the rebuttal will be sufficient.
Author Response
Reply: We would like to thank Reviewer 3 for the comment. We have replaced Fig. 1A with the one with N2 as suggested. We deeply appreciate Reviewer 3 for all the constructive comments and suggestions in the manuscript revision to substantially improve the quality.

Round 3
Reviewer 3 Report
The authors addressed all the comments and I support acceptance of the manuscript.
Author Response
We would like to thank Reviewer 3 for the support as well as all the constructive comments and suggestions to substantially improve the quality of our revised manuscript. Merry Christmas and Happy New Year!
